# The Influence of Oxygen Activity on Phase Composition, Crystal Structure, and Electrical Conductivity of CaV$_{1-x}$Mo$_x$O$_{3\pm\delta}$

**Semyon A. Belyakov** [1,*], **Evgeny Yu. Gerasimov** [2] and **Anton V. Kuzmin** [3]

[1] Laboratory of Electrochemical Materials Science, Institute of High-Temperature Electrochemistry UB RAS, 620137 Yekaterinburg, Russia
[2] Federal Research Center Boreskov Institute of Catalysis SB RAS, 630090 Novosibirsk, Russia; gerasimov@catalysis.ru
[3] Department of Technology of Inorganic Materials and Electrochemical Production, Vyatka State University, 610000 Kirov, Russia; a.v.kuzmin@yandex.ru
[*] Correspondence: bca2@mail.ru

**Abstract:** Perovskite–like vanadate–molybdates are interesting from the point of view of their metal–like conductivity, which combines the correlated and free electron nature. A series of CaV$_{1-x}$Mo$_x$O$_{3-\delta}$ solid solutions was considered near the Mo concentration x = 0.4, where a difficult–to–perceive structural transition was previously detected. High-resolution transmission electron microscopy revealed the phase separation of CaV$_{0.6}$Mo$_{0.4}$O$_{3-\delta}$ into nanoscale regions with different ratios of V and Mo concentrations, despite X–ray diffraction analysis exhibiting a homogeneous perovskite structure. The rest of the compositions from the CaV$_{1-x}$Mo$_x$O$_{3-\delta}$ series do not show phase separation. The nonmonotonic behavior of the conductivity and linear expansion of CaV$_{1-x}$Mo$_x$O$_{3\pm\delta}$ was shown when the oxygen activity in the N$_2$-H$_2$-H$_2$O gas mixture was varied, which is mainly determined by the partial decomposition of the perovskite phase. Against this background, the behavior of the electrical properties of the CaV$_{1-x}$Mo$_x$O$_{3\pm\delta}$ individual phase remains unclear.

**Keywords:** perovskite; vanadate; molybdate; metal–like conductivity; chemical expansion; phase separation

## 1. Introduction

Perovskite–like vanadates ((Ca,Sr)VO$_3$) and molybdates ((Ca,Sr)MoO$_3$) represent an interesting class of materials and are oxides with metal–like conductivity. Overlapping between neighboring d–orbitals of V or Mo through oxygen 2p–orbitals leads to a strong delocalization of electrons, which is a source of tunneling electron transfer [1,2]. Orthorhombic distortions of the crystal structure in CaVO$_3$ and CaMoO$_3$ reduce the Me–O–Me bond angle to 160° [3,4], relative to the ideal 180° in cubic SrVO$_3$ and SrMoO$_3$, which is accompanied by a decrease in the orbital overlap integral and, therefore, leads to a decrease in the transfer electron bandwidth [1,5,6]. A similar phenomenon is observed for vanadate–molybdates ((Ca,Sr)V$_{0.5}$Mo$_{0.5}$O$_3$) [7].

Vanadates are correlated metals, as they exhibit strong electron–electron interactions [2,8], which opens up a number of interesting uses. Vanadium oxides are considered as promising systems, and undergo a metal–insulator transition (Mott transition) [9] which makes their use in electronics possible, for example, in components of transistors or relaxation oscillators. However, the metal–insulator transition, although expected, was not previously observed in the bulk form of (Ca,Sr)VO$_3$ [8,10]. It was shown that dimensional constraints, when in the form of thin films, and epitaxial deformation enhance the correlation effects, which brings these systems closer to the metal–insulator transition [11–13]. In addition, a new approach to the design of oxide transparent conductors is based on controlling the concentration and effective mass of delocalized electrons, in order to achieve high

conductivity and transparency in the visible region of the electromagnetic spectrum [10,14]. This allows metal–like oxides to be used as transparent electrodes in organic light–emitting diodes. Finally, high electronic conductivity combined with an oxygen transfer at high temperatures makes it possible to use partially substituted vanadates as a fuel electrode in electrochemical power sources [15–17], although their low thermodynamic stability at high temperatures remains a problem [18,19].

Molybdates, unlike vanadates, do not exhibit electron–electron scattering, and are similar in behavior to normal metals [1,20,21]. Their use as electrodes in electrochemical devices is limited due to their low redox stability [22,23], although with partial substitution of Mo for strontium ferrites, promising results can be obtained [24,25], but with a significant decrease in the conductivity of the electrode material.

The effect of oxygen nonstoichiometry ($\delta$) on the structure and electronic properties of $CaVO_{3-\delta}$ was studied by Ueda et al. [26,27]. The initially detected transition from a metallic state to an insulator [28] was associated with a partial oxidation of the ceramic surface [26,29]. A small region of oxygen over–stoichiometry $(3 + \delta)$ in $(Ca,Sr)VO_{3\pm\delta}$ vanadates was shown [30,31], after which oxidation to the phases $(Ca,Sr)_3V_2O_8$ and $(Ca,Sr)_2V_2O_7$ was observed [15,16,30]. On the other hand, the use of highly reducing conditions for the synthesis of perovskite–like vanadates implies a large oxygen deficiency. In perovskite–like oxides, a transition to domainized brownmillerite–like structures is possible, as is well–known for strontium ferrite [32], although additions of V and Mo to strontium ferrite "smooth out" these transitions and reduce domain wall thickness [33–35]. The perovskite–brownmillerite transition may be preceded by the ordering of oxygen vacancies. Accordingly, Ueda et al. [26,27] showed the formation of oxygen vacancy channels for $CaVO_{3-\delta}$ at different values of oxygen nonstoichiometry. It should be taken into account that even small structural distortions have a large effect on the electronic structure of perovskite–like vanadates. The Fermi surface shape for $CaVO_3$ was investigated by Inoue et al. [36] in the form of an ideal model only, which does not reflect the effect of oxygen nonstoichiometry and the related distortions of the crystal structure of the material. The effect of oxygen activity ($aO_2$) on the electronic structure of vanadates and molybdates has scarcely been studied. The significant influence of oxygen vacancies has been shown only on the electronic structure of vanadates [37], while in molybdates their influence is considered insignificant [21].

Perovskite–like vanadate–molybdates $(Ca,Sr)(V,Mo)O_{3-\delta}$ are interesting from the point of view of their electronic properties, as they combine the correlated electron nature of vanadates and the free electron nature of molybdates [38]. Unfortunately, the effect of oxygen activity on the properties of vanadate–molybdates has not yet been studied. Earlier, we presented information on the behavior of the electronic properties with temperature of $CaV_{1-x}Mo_xO_{3-\delta}$ ($0 \le x \le 0.6$) in a dry hydrogen atmosphere [38]. In addition, we were puzzled by the previously discovered transition region at the Mo concentration $x = 0.4$ [38,39], where some structurally related changes take place. In the current work, we examine in detail the structure and microstructure of powders and ceramics of $CaV_{1-x}Mo_xO_{3-\delta}$ ($0.3 \le x \le 0.5$), and evaluate the effect of oxygen activity on conductivity and chemical expansion.

## 2. Materials and Methods

The $CaV_{1-x}Mo_xO_{3-\delta}$ ($0.3 \le x \le 0.5$) powders were prepared via pyrolysis of formate solutions according to the technique from [40]. Stoichiometric quantities of $CaCO_3$, $NH_4VO_3$ and $(NH_4)_6Mo_7O_{24}\cdot4H_2O$ were dissolved in formic acid. The obtained solutions were evaporated and dried at 473 K in air. The resulting powders were sintered gradually at 873 K for 6 h in air to remove organic components. To obtain single–phase materials, a two–stage annealing process was performed, at 1173 and 1473 K for 3 h in a $H_2$-$H_2O$(3%) gas mixture. The elemental composition of the prepared samples was characterized by atomic–emission spectroscopy (AES) with plasma analysis, using a Perkin Elmer Optima 4300 DV spectrometer (PerkinElmer, Waltham, MA, USA). Powders were pressed at 400 MPa into

bulk ceramic samples and formed by annealing at 1473 K for 3 h in a $H_2$-$H_2O$(3%) gas mixture. The heating and cooling rate was 3 K min$^{-1}$.

The phase composition and structure of as–prepared powders and ceramics were studied at room temperature via X–ray diffraction (XRD) analysis, using a Rigaku DMAX diffractometer (Rigaku, Tokyo, Japan) with Cu$_{K\alpha}$ radiation with step $2\theta = 0.02°$. The phase analysis was based on the PDF-2 database. Unit cell parameters were calculated based on interplanar distances using CellRef v.3.0 software. The morphology of ceramic samples was investigated via scanning electron microscopy (SEM) in a Tescan MIRA 3 LMU microscope (Tescan, Brno, Czech Republic) using secondary electron (SE) and back–scattered electron (BSE) operating modes. The original surface of ceramic samples was examined without pretreatment or etching. The maps of elemental distributions were obtained via energy–dispersive X–ray spectroscopy (EDX), using an Oxford Instruments X-Max 80 INCA Energy 350 microanalysis system with a non–nitrogen detector (Oxford Instruments, Abingdon, UK). Measurements of the relative density of the ceramic samples were carried out according to the Archimedes method, by weighing them in kerosene on CAS CAUX 220 analytical scales (CAS, Seoul, South Korea). High–resolution transmission electron microscopy (HRTEM) analysis was used for the refinement of microstructure of powder samples. HRTEM analysis was performed using JEM-2010 equipment (JEOL, Tokyo, Japan), operating at a line resolution of 0.14 nm. Microanalysis of elemental composition by EDX analysis (XFlash, Bruker, Billerica, MA, USA) was used, with an Si detector at 127 eV resolution. For preparation of the samples for HRTEM observation, a drop of the specimen in suspension, ultrasonically treated in ethanol, was placed into a holey–carbon film supported on Cu grids.

The measurements of electrical conductivity, Seebeck coefficient and linear expansion were performed at 1073 K in a $N_2$-$H_2$-$H_2O$ gas mixture with an $a$O$_2$ range of $10^{-21}$ to $10^{-15}$ atm. The composition of the $N_2$-$H_2$-$H_2O$ gas mixture was set by an electrochemical pump based on an yttria–stabilized zirconia (YSZ) ceramic test tube. The oxygen activity ($a$O$_2$) was controlled using an electrochemical sensor, also based on YSZ. Humidity was set by bubbling of gas mixture through water at room temperature. Electrical conductivity measurements were performed by 4–probe DC method using a Hioki RM-3542 microohmmeter (Hioki, Nagano, Japan). The samples were rectangular bars with four Pt electrodes, made of dispersed Pt powder and sintered at 1373 K for 1 h in a $N_2$-$H_2$-$H_2O$ gas mixture. Seebeck coefficients were measured using a state–of–the–art setup based on L-card 24-bit ADC. Several open circuit voltage (OCV) values were collected at a series of temperature gradients ($\Delta T = 0$–15 K) by means of a built–in micro heater in the measuring cell. The Seebeck coefficient was calculated by the slope of the OCV = f($\Delta T$) dependency. The Seebeck coefficient of $-18.25$ $\mu$V K$^{-1}$ for Pt wires at 1073 K [41] was taken into account. The linear expansion of ceramics with atmospheric composition (chemical expansion) was investigated using dilatometry. A quartz dilatometer based on TESA Tesatronic TT-80 equipment with a GT-21HP probe (TESA, Renens, Switzerland) was used. The sample length at 1073 K and $a$O$_2 = 10^{-19.5}$ atm (corresponding to the value of OCV = 1 V on the oxygen electrochemical sensor) was taken as the reference point for the relative expansion.

## 3. Results and Discussion

### 3.1. Materials Characterization

Figure 1 shows XRD patterns of as–prepared CaV$_{1-x}$Mo$_x$O$_{3-\delta}$ ($0.3 \leq x \leq 0.5$) samples. A low content of metallic Mo was found in some samples. Also possible are minor impurities of CaO, monoclinic Ca$_3$V$_2$O$_8$ or Ca$_2$V$_2$O$_7$, which are beyond the XRD detection line. In general, the chemical composition of the prepared samples conformed to initial reagent concentration, according to AES analysis.

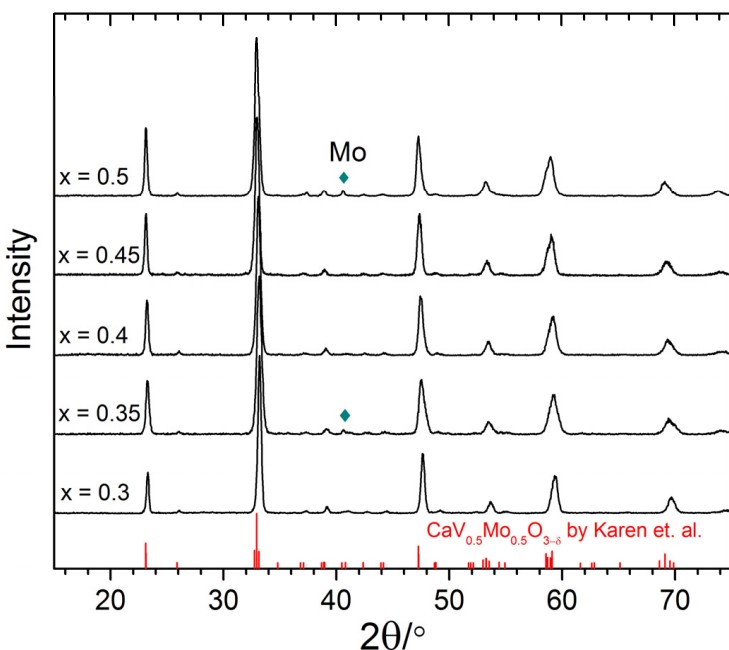

**Figure 1.** XRD patterns of as–prepared $CaV_{1-x}Mo_xO_{3-\delta}$ ($0.3 \leq x \leq 0.5$) powders.

The crystal structure of $CaV_{1-x}Mo_xO_{3-\delta}$ solid solutions is described within the orthorhombic perovskite lattice with the *Pnma* space group, as reported for $CaVO_{3-\delta}$ [3] and $CaMoO_{3-\delta}$ [4], as well as for $CaV_{0.5}Mo_{0.5}O_{3-\delta}$ [7,42]. With an increase in the Mo concentration in $CaV_{1-x}Mo_xO_{3-\delta}$, the XRD peaks shift to smaller angles, indicating an increase in the unit cell parameters as the ionic radii of Mo are larger than V (for the average 4+ state). Figure 2 presents the calculated unit cell parameters of the orthorhombic cell of $CaV_{1-x}Mo_xO_{3-\delta}$. A bend on the dependence of the *b* and *c* unit cell parameters occurs, whereas the *a* parameter behaves linearly. A similar effect in the form of a structural transition from cubic to tetragonal syngony in $SrFe_{1-x}Mo_xO_{3-\delta}$ at the Mo concentration $x = 0.4$ was found in [43]. Subsequently, the authors of [44] found that, with an increase in the Mo concentration in $SrFe_{1-x}Mo_xO_{3-\delta}$, the fraction of ordered regions of the double perovskite $Sr_2FeMoO_6$ gradually increases. Merkulov et al. [45] confirmed the ordering of Fe and Mo at the level of nanosized domains as the double perovskite $Sr_2FeMoO_6$, within a disordered matrix of the orthorhombic perovskite of $SrFe_{0.7}Mo_{0.3}O_{3-\delta}$. However, Karen et al. [7] showed, via synchrotron X–ray powder diffraction analysis, the absence of ordering of V and Mo in the B site of $CaV_{0.5}Mo_{0.5}O_{3-\delta}$, thus the formation of a double perovskite structure $Ca_2VMoO_6$ is extremely unlikely. Therefore, XRD analysis does not provide an understanding of the structural differences in a number of $CaV_{1-x}Mo_xO_{3-\delta}$ ($0.3 \leq x \leq 0.5$) solid solutions.

Figure S1 in the Supplementary Materials shows SEM images (SE mode) of sintered $CaV_{1-x}Mo_xO_{3-\delta}$ ($0.3 \leq x \leq 0.5$) ceramic samples. A high porosity of about 35% can be observed, which is also confirmed by measurements via the Archimedes method. A high porosity of ceramics is characteristic of such compounds [7,17]. Most of the ceramic grains under study are approximately a few μm in size, with many grains forming large agglomerates measuring 5–10 μm in size. Samples $CaV_{0.7}Mo_{0.3}O_{3-\delta}$ and $CaV_{0.5}Mo_{0.5}O_{3-\delta}$ show a wide variation in grain sizes, while $CaV_{0.6}Mo_{0.4}O_{3-\delta}$ shows a more even distribution of grain sizes.

The results from a more detailed SEM study of ceramics, with the use of the BSE mode, are shown in Figure 3. Samples $CaV_{0.7}Mo_{0.3}O_{3-\delta}$ and $CaV_{0.5}Mo_{0.5}O_{3-\delta}$ shows a homogeneous image, with the exception of small light grains that are impurities of metallic Mo, as recorded on the XRD diagrams. On the contrary, sample $CaV_{0.6}Mo_{0.4}O_{3-\delta}$ demonstrates contrast, indicating a difference in the chemical composition of individual

grains. Unfortunately, the complex relief of the samples does make it possible to obtain an adequate EDX analysis of elemental concentrations.

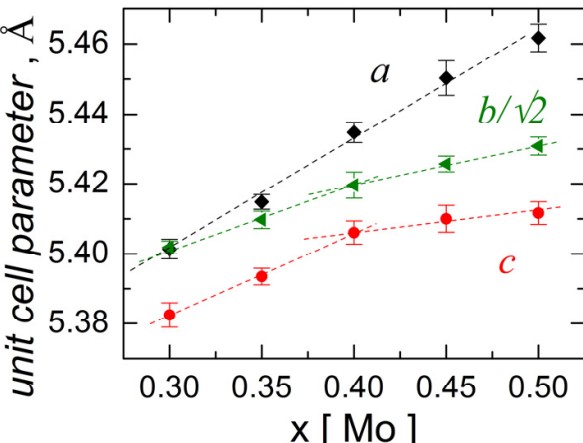

**Figure 2.** The dependence of *a*, *b* and *c* unit cell parameters (*Pnma* space group) of $CaV_{1-x}Mo_xO_{3-\delta}$ on Mo concentration.

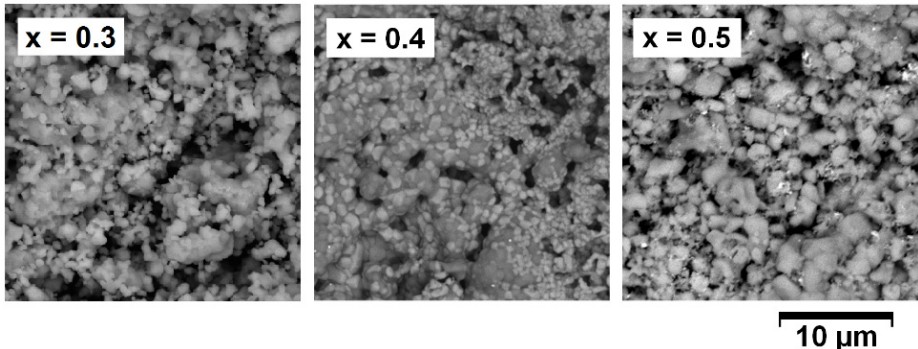

**Figure 3.** SEM images (BSE mode) of the surface of $CaV_{1-x}Mo_xO_{3-\delta}$ (x = 0.3, 0.4, 0.5) ceramic samples.

A detailed study of the $CaV_{0.6}Mo_{0.4}O_{3-\delta}$ and $CaV_{0.5}Mo_{0.5}O_{3-\delta}$ powders was performed via HRTEM. The $CaV_{0.5}Mo_{0.5}O_{3-\delta}$ powder sample is present as well crystallized particles of the perovskite phase, with a size of about 1 μm. The interplanar distance of 2.165 Å observed in Figure 4a corresponds to the {022} set for the orthorhombic perovskite lattice. For convenience, we have provided real resolution HRTEM images in the Supplementary Materials. The EDX data on the surface of the observed particles, in general, corresponds to the chemical composition established during the synthesis. However, it should be noted that regions with an increased content of Ca cations are recorded, which corresponds to the enrichment of the perovskite surface with calcium oxide. Previously, Ca segregation was detected via X–ray photoelectron spectroscopy [39]. The Cu reflex on the EDX spectrum is due to the Cu mesh that was used as a powder holder.

Figure 4b shows the HRTEM results for the $CaV_{0.6}Mo_{0.4}O_{3-\delta}$ powder sample, which demonstrates a large scattering in crystallite sizes, from 50 nm to 500 nm. According to EDX data, crystallites have different chemical compositions; firstly, the contents of V and Mo differ. Three characteristic regions can be found: enriched in Mo (region 1); with an equal ratio of V and Mo (region 2); or enriched in V (region 3). In all cases, the sets of interplanar spacings for $CaV_{0.6}Mo_{0.4}O_{3-\delta}$ are well-described within the orthorhombic perovskite lattice, which is also the case for the $CaV_{0.5}Mo_{0.5}O_{3-\delta}$ sample. Thus, the $CaV_{0.6}Mo_{0.4}O_{3-\delta}$ sample is characterized by phase separation into nanosized regions with different cationic compositions, which have a similar crystal structure. This fact explains the observed single–phase state of $CaV_{0.6}Mo_{0.4}O_{3-\delta}$, according to XRD (Figure 1), which does not show significant

peak broadening. The phase decomposition often occurs in solid solutions with limited component solubility [46], as well as separation of the main crystal lattice into two or more lattices. For example, such cases of phase separation are observed in disordered perovskite phases, in the form of partial ordering in nanoscale domains with a double perovskite structure [45,47]. However, almost nothing is known about phase separation into completely isostructural phases. It is possible that this phenomenon is caused by a peculiar distribution of the charge states of V and Mo, by analogy with the effect of Re in solid solutions of $V_{1-x}Re_xO_2$ with a rutile structure [48]. Previously, we observed an anomalous distribution of the charge states of $V^{3+/4+}$ in $CaV_{0.6}Mo_{0.4}O_{3-\delta}$, according to X–ray photoelectron spectroscopy analysis [39]. The most surprising fact is that the phase separation of the material occurs within a very limited region of $CaV_{1-x}Mo_xO_{3-\delta}$ solid solutions, near the Mo concentration x = 0.4. Further structural clarifications are necessary in order to understand how $CaV_{1-x}Mo_xO_{3-\delta}$ solid solutions differ above and below x = 0.4.

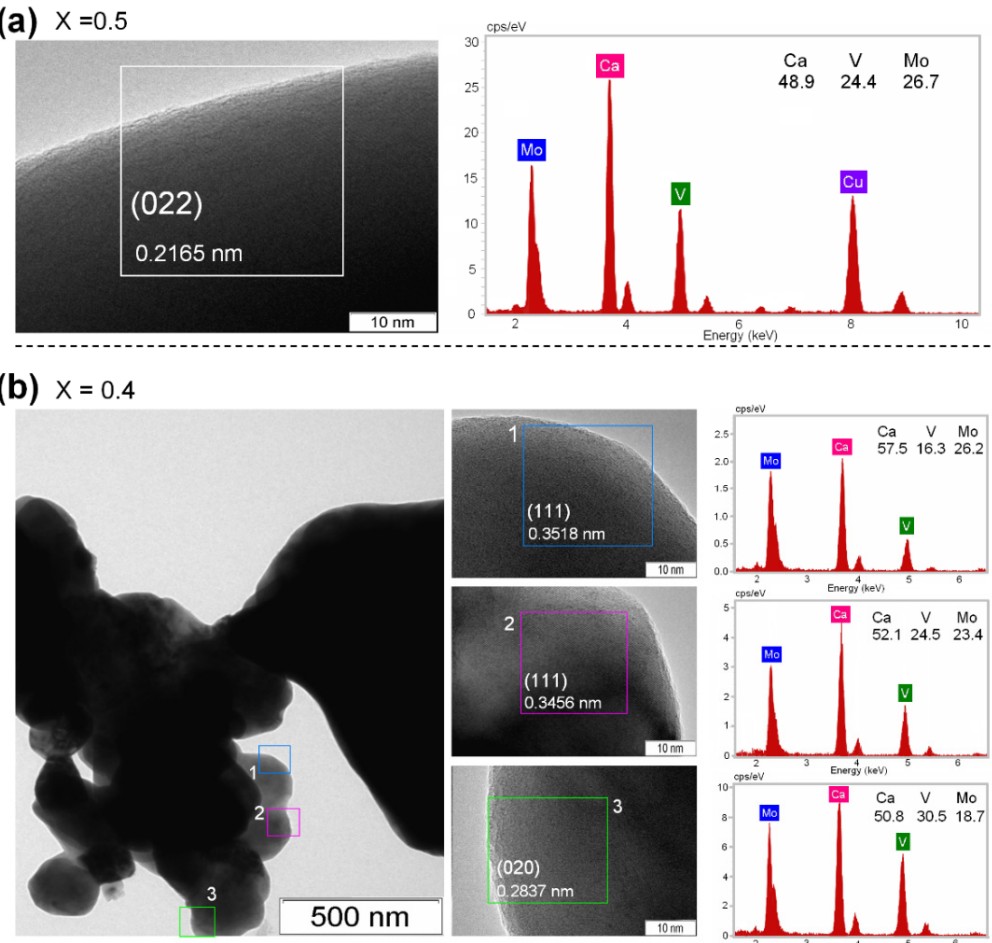

**Figure 4.** HRTEM images of $CaV_{0.5}Mo_{0.5}O_{3-\delta}$ (**a**) and $CaV_{0.6}Mo_{0.4}O_{3-\delta}$ (**b**) powders with results of EDX elemental analysis.

### 3.2. Impact of $aO_2$ on Properties of the Materials

The main course of measurements of electrical conductivity and chemical expansion was carried out under highly reducing conditions, at $aO_2 = 10^{-21}$ atm with a gradual increase in oxygen activity up to $aO_2 = 10^{-15}$ atm with reverse for checking thermodynamic equilibrium. We observed a long equilibration of approximately 20 h for each experimental point. Taking into account the temperature of 1073 K and the high porosity of the samples, it is difficult to connect the above fact with the diffusion of oxygen through the sample. Most likely, the long time taken in reaching equilibrium is due to the diffusion of cations.

For example, a change in the mutual ordering of V and Mo can be expected, as earlier studies [7] showed the absence of long-range order between V and Mo in the crystal lattice of $CaV_{0.5}Mo_{0.5}O_{3-\delta}$.

Figure 5a shows the conductivity dependencies on $aO_2$ of $CaV_{1-x}Mo_xO_{3-\delta}$ ($0.3 \leq x \leq 0.5$) ceramic samples at 1073 K. The dependences are nonmonotonic and have a maximum in the region of $aO_2 = 10^{-19}$ atm. At the same time, the Seebeck coefficient for $CaV_{1-x}Mo_xO_{3-\delta}$ at 1073 K is nearly independent of $aO_2$ (Figure 5b). As the main charge carriers in $CaV_{1-x}Mo_xO_{3-\delta}$ are delocalized electrons [7,38], the change in the conductivity of materials with varying $aO_2$ should primarily be associated with their concentration and mobility. The electron mobility depends on the type of electron scattering. Earlier, we showed the mixed character of electron scattering in $CaV_{1-x}Mo_xO_{3-\delta}$, which is intermediate between electron–phonon and electron–electron [38].

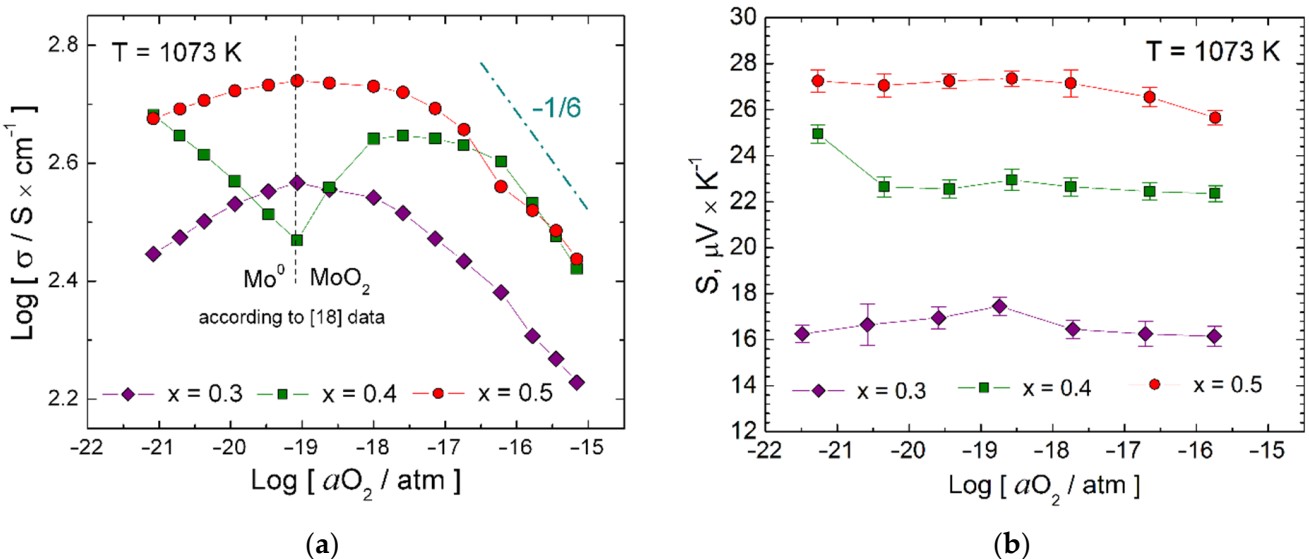

**Figure 5.** The dependencies of conductivity (**a**) and Seebeck coefficient (**b**) of $CaV_{1-x}Mo_xO_{3\pm\delta}$ ($0.3 \leq x \leq 0.5$) ceramics on $aO_2$ at 1073 K.

It is known that in conventional metallic conductors, the concentration of free electrons is nearly independent of external conditions. At the same time, for oxide materials in the region of small $aO_2$ values, Brower's model assumes the formation of oxygen vacancies ($V_O^{\bullet\bullet}$) from oxygen in crystal lattice sites ($O_O^{\times}$) with the generation of two n–type electrons ($e^-$) per vacancy, according to the equation

$$O_O^{\times} = V_O^{\bullet\bullet} + \frac{1}{2}O_2 + 2e^-. \tag{1}$$

In this case, the slope of the logarithmic dependence of the concentration of electronic defects on $aO_2$ will correspond to –1/6, in the case of the electronic type of disordering [49]. In Figure 5a, it can be seen that the slope of the logarithmic dependences of conductivity in the $aO_2$ region from $10^{-15}$–$10^{-19}$ atm is less than the expected value (–1/6), which might be associated with a decrease in the electron mobility. The authors of [29] came to a similar conclusion when studying the effect of oxygen nonstoichiometry on the conductivity of $CaVO_{3-\delta}$, assuming additional scattering of electrons by oxygen vacancies. Likewise, a similar suggestion was given in [50] while studying the conductivity of $SrV_{1-x}Nb_xO_{3-\delta}$ systems. The effect of oxygen vacancies on the electronic structure of $SrVO_3$ was recently discussed in [37], where it was demonstrated that the shape of the Mott–Hubbard band changes under the influence of created oxygen vacancies.

The presence of a clear maximum of conductivity at $aO_2 = 10^{-19}$ atm (Figure 5a), with a subsequent decrease in conductivity at lower $aO_2$, is difficult to correlate with the influence of one type of electron scattering. Similar dependencies of conductivity with

a maximum were observed by Macias et al. for $SrVO_{3-\delta}$ [16] and substituted strontium vanadates [15,17,50], where the behavior of conductivity under highly reducing conditions was explained by the transition of vanadium from $V^{3+}$ to $V^{4+}$. In addition, the observed phenomenon is similar to the data reported by Hui and Petric [18] for the $MoO_2$–Mo transition at 1073 K (dashed line in Figure 5a). In the region of $aO_2$ less than $10^{-19}$ atm, the behavior of $CaV_{1-x}Mo_xO_{3-\delta}$ can be described by the equilibrium of the main perovskite phase with the phase of metallic Mo. This is confirmed by the XRD pattern of $CaV_{0.5}Mo_{0.5}O_{3-\delta}$ (Figure 6) obtained after holding for 54 h at $aO_2 = 10^{-21}$ atm, which shows an admixture of metallic Mo. At the same time, after holding at $aO_2 = 10^{-19}$ atm, we can observe the disappearance of the Mo impurity and the appearance of XRD reflections of the tetragonal phase of $CaMoO_4$. After holding at $aO_2 = 10^{-17}$ atm, the concentration of the $CaMoO_4$ tetragonal phase increases. The high metal–like conductivity of the material remains up to $aO_2 = 10^{-15}$ atm. A long process of the degradation of electrical conductivity of $CaV_{0.5}Mo_{0.5}O_{3\pm\delta}$ begins at $aO_2 > 10^{-15}$ atm. Figure 6 shows XRD patterns of the surface of the ceramic sample after 54 h of exposure at $aO_2 = 10^{-14.7}$ atm, where the complete decomposition of the perovskite–like phase into $CaMoO_4$ with a tetragonal structure and $Ca_3V_2O_8$ with a monoclinic structure can be observed. Thus, the decrease in the conductivity of $CaV_{0.5}Mo_{0.5}O_{3\pm\delta}$ in the region near $aO_2 > 10^{-19}$ atm occurs due to the progressive decomposition of the metal–like phase into two insulating phases, which is clearly seen from the evolution of XRD patterns in Figure 6. For comparison, an XRD pattern of the $CaV_{0.5}Mo_{0.5}O_{3-\delta}$ powder after oxidation in air at 1473 K is shown, in which only the calcium molybdate phase with a tetragonal structure remains; whereas, $Ca_3V_2O_8$ with a monoclinic structure can transform into $Ca_2V_2O_7$ with a pyrochlore structure, which, at temperatures above 1273 K, transforms into an amorphous form [42]. Figure 7 shows SEM images of the surfaces of oxidized samples. One can see the higher density of the oxidized ceramics relative to the originals, which indicates sintering during the decomposition of the perovskite phase. Unfortunately, the resolution of the EDX detector does not allow us to estimate the distribution of elements, although some uneven distribution of V and Mo is observed for sample $CaV_{0.5}Mo_{0.5}O_{3\pm\delta}$. The $aO_2$–boundary for $CaV_{0.5}Mo_{0.5}O_{3\pm\delta}$, where high metal–like conductivity is found, can be considered as the $aO_2$ region near approximately $10^{-16}$ atm, which is in agreement with [42], and is also in good agreement with [16] for $SrVO_{3-\delta}$.

Figure 8 demonstrates chemical expansion and electrical conductivity dependencies on $aO_2$ for $CaV_{1-x}Mo_xO_{3-\delta}$ (x = 0.4 and 0.5) at 1073 K. An increase in $aO_2$ from $10^{-21}$ up to $10^{-19}$ atm causes a linear compression of the $CaV_{0.5}Mo_{0.5}O_{3-\delta}$ ceramic sample, due to a decrease in the ionic radii of V and Mo with an increase in their oxidation state, which is consistent with conventional concepts [51,52]. In the region of $aO_2 = 10^{-17.5}$ atm, a minimum of chemical expansion can be observed, after which the sample begins to expand up to $aO_2 = 10^{-16}$ atm, and finally an intensive expansion of the sample occurs. This is due to the partial decomposition of $CaV_{0.5}Mo_{0.5}O_{3-\delta}$ according to the XRD results (Figure 6). Figure 8b shows the dependencies of conductivity and linear expansion of $CaV_{0.6}Mo_{0.4}O_{3-\delta}$ on $aO_2$ at 1073 K, which differ significantly from the behavior of the $CaV_{0.5}Mo_{0.5}O_{3-\delta}$ sample. The conductivity of $CaV_{0.6}Mo_{0.4}O_{3-\delta}$ shows nonmonotonic changes at $aO_2 < 10^{-18}$ atm and has a local minimum at $aO_2 = 10^{-19}$ atm. The chemical expansion of the $CaV_{0.6}Mo_{0.4}O_{3-\delta}$ ceramic sample demonstrates a difficult–to–explain relationship that does not agree with conventional concepts. Taking into account the phase separation in the $CaV_{0.6}Mo_{0.4}O_{3-\delta}$ sample into nanoscale regions with different chemical compositions, as detected via HRTEM, it can be assumed that the defect structure of the material will show a composite nature.

Figure 9a demonstrates the long–term relaxation of the chemical expansion of $CaV_{1-x}Mo_xO_{3-\delta}$ over several tens of hours. Upon a change in $aO_2$ from $10^{-16.7}$ to $10^{-16.2}$ atm for the $CaV_{0.5}Mo_{0.5}O_{3\pm\delta}$ sample and from $10^{-15.8}$ to $10^{-15.3}$ atm for the $CaV_{0.6}Mo_{0.4}O_{3\pm\delta}$ sample there is a long process of expansion, during which there is a significant change in its linear dimensions. The absorption of oxygen is accompanied by the formation of monoclinic

Ca$_3$V$_2$O$_8$ and tetragonal CaMoO$_4$ phases from perovskite–like CaV$_{1-x}$Mo$_x$O$_{3-\delta}$, which leads to intense chemical expansion. A small difference in the $a$O$_2$ values is associated with different concentrations of the V–Mo pair. The mechanism of kinetics of chemical expansion relaxation was investigated using a dependency approach of degree of transformation (($\Delta$L/L)$_{normalized}$) on reduced time (t/t$_{0.5}$) [53]. Several basic mechanisms of solid–state reactions were considered: diffusion of components controlled reaction, solid–state reaction following first order kinetics, phase boundary controlled reaction, and Avrami–Erofeev kinetic equation under limitations, due to the nucleation stage of the reaction. The form of typical dependences of ($\Delta$L/L)$_{normalized}$ = f(t/t$_{0.5}$) for various mechanisms of solid–phase reactions with experimental data is shown in Figure 9b. It can be concluded that the closest correspondence to the relaxation dependence of the chemical expansion of CaV$_{1-x}$Mo$_x$O$_{3\pm\delta}$ samples is related to the solid–phase reaction mechanism, in good agreement with the observed decomposition of the perovskite–like CaV$_{0.5}$Mo$_{0.5}$O$_{3-\delta}$ into tetragonal CaMoO$_4$ and monoclinic Ca$_3$V$_2$O$_8$.

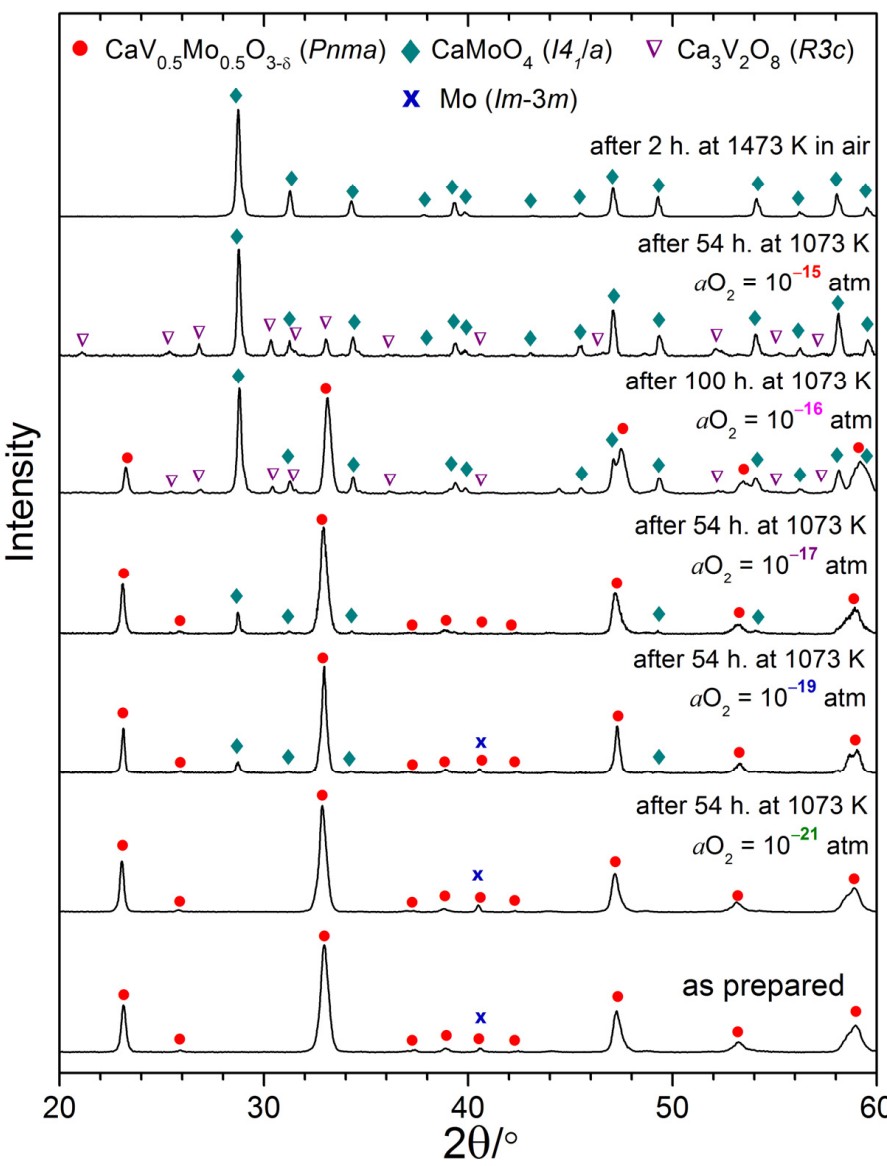

**Figure 6.** XRD patterns of CaV$_{0.5}$Mo$_{0.5}$O$_{3\pm\delta}$ samples after an exhibition at various conditions.

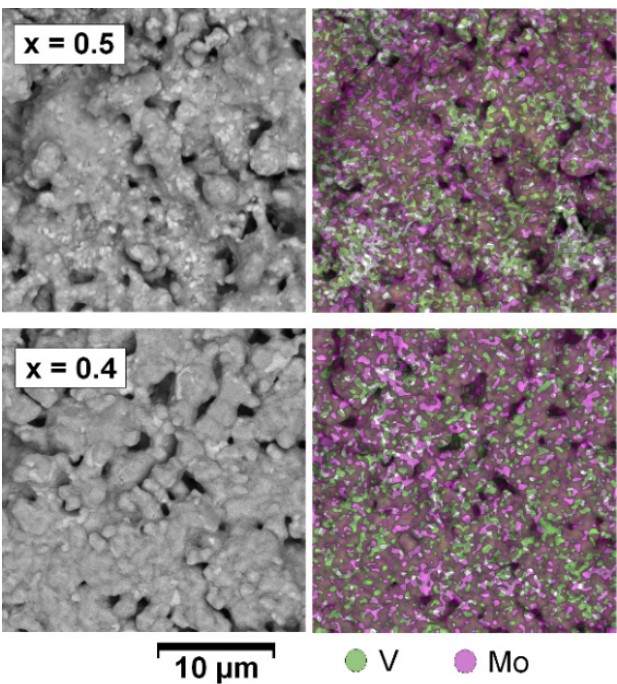

**Figure 7.** SEM images (BSE mode) of surface of $CaV_{1-x}Mo_xO_{3\pm\delta}$ (x = 0.4 and 0.5) ceramic samples after oxidation at 1073 K and $aO_2 = 10^{-15}$ atm with EDX maps of V and Mo distribution.

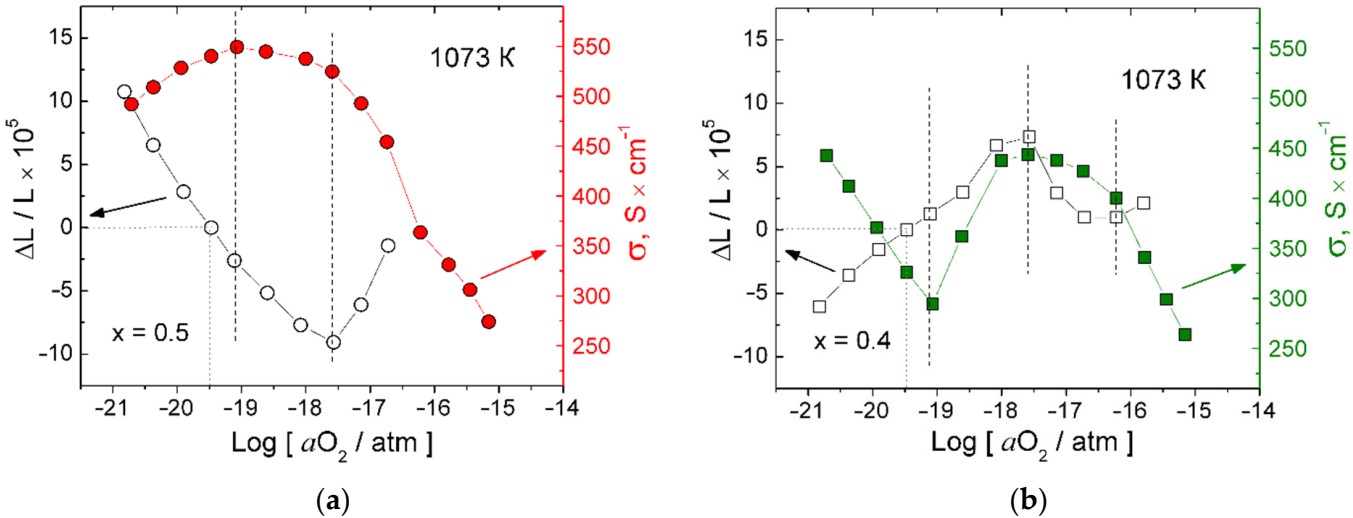

(**a**)                            (**b**)

**Figure 8.** The dependencies of chemical expansion (left axes) and conductivity (right axes) on $aO_2$ at 1073 K for: $CaV_{0.5}Mo_{0.5}O_{3\pm\delta}$ (**a**) and $CaV_{0.6}Mo_{0.4}O_{3\pm\delta}$ (**b**).

The behavior of the electrical properties of the individual $CaV_{1-x}Mo_xO_{3\pm\delta}$ perovskite phases from $aO_2$ remains poorly understood against the background of the phase decomposition. The positive sign of the Seebeck coefficient (Figure 5b) indicates that the delocalized electrons in $CaV_{1-x}Mo_xO_{3\pm\delta}$ are p–type electrons (electronic holes) in the entire studied area of $aO_2$. In addition, small changes in the conductivity and Seebeck coefficient indicate a small effect of $aO_2$ on the concentration and mobility of delocalized p–type electrons in $CaV_{1-x}Mo_xO_{3\pm\delta}$, although the opposite was expected. Apparently, $CaV_{1-x}Mo_xO_{3\pm\delta}$ has a very small region of oxygen nonstoichiometry, which causes its progressive decomposition with an increase or decrease in $aO_2$. Accordingly, significant changes in the crystal and electronic structure of the $CaV_{1-x}Mo_xO_{3\pm\delta}$ perovskite phase itself are not expected.

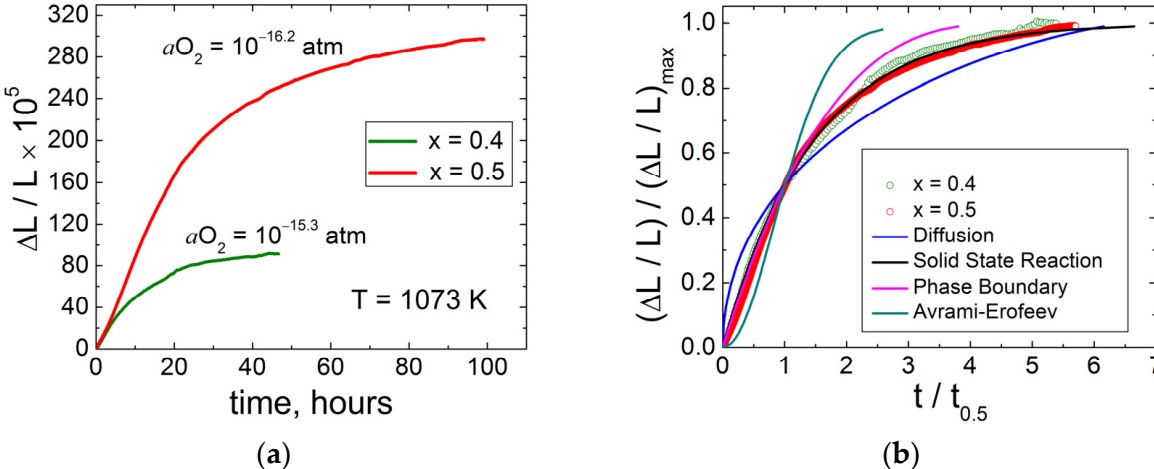

**Figure 9.** The results of chemical expansion study of oxidation process in $CaV_{1-x}Mo_xO_{3\pm\delta}$ (x = 0.4 and 0.5) at 1073 K and constant $aO_2$ value in $\Delta L/L = f(t)$ (**a**) and $(\Delta L/L)_{normalized} = f(t/t_{0.5})$ (**b**) plots.

## 4. Conclusions

Solid solutions $CaV_{1-x}Mo_xO_{3-\delta}$ with an orthorhombic perovskite structure near the structural transition at x = 0.4 were considered, and showed a bend in the dependencies of *b* and *c* unit cell parameters on the Mo concentration. The phase separation of $CaV_{0.6}Mo_{0.4}O_{3-\delta}$ into nanoscale regions with different contents of V and Mo was observed. It is noteworthy that phase separation occurs within a narrow range of V and Mo concentrations. We see no reason to classify the regions of $CaV_{1-x}Mo_xO_{3-\delta}$ solid solutions before and after x = 0.4 as fundamentally different types of solid solutions, although further structural refinements are needed.

The article considers the behaviors of conductivity and chemical expansion of $CaV_{1-x}Mo_xO_{3\pm\delta}$ at 1073 K depending on the activity of oxygen in the gas phase ($aO_2$), under the reducing conditions of a $N_2$-$H_2$-$H_2O$ gas mixture. Both types of dependencies exhibit nonmonotonic behavior. Despite the expected significant effect of $aO_2$ on the concentration and mobility of delocalized p–type electrons in $CaV_{1-x}Mo_xO_{3\pm\delta}$, the phase decomposition causes major changes in electrical conductivity and chemical expansion of the materials under study. In the reduction region at $aO_2$ less than $10^{-19}$ atm, there is a phase equilibrium with metallic Mo, while the oxidation of materials at $aO_2 > 10^{-19}$ atm is accompanied by the progressive decomposition of the metal–like perovskite V–Mo phase into insulating tetragonal calcium molybdate and monoclinic calcium vanadate, and at $aO_2 = 10^{-15}$ atm, complete decomposition is observed.

**Supplementary Materials:** The following supporting information can be downloaded at: https://www.mdpi.com/article/10.3390/cryst12030419/s1. Figure S1: SEM images (SE mode) of the surface of sintered $CaV_{1-x}Mo_xO_{3-\delta}$ ($0.3 \leq x \leq 0.5$) ceramic samples; Figure S2: Real–resolution HRTEM image supported by FFT diffraction patterns for $CaV_{0.5}Mo_{0.5}O_{3-\delta}$ sample; Figure S3: Real–resolution HRTEM image supported by FFT diffraction patterns for $CaV_{0.6}Mo_{0.4}O_{3-\delta}$ sample (region 1); Figure S4: Real–resolution HRTEM image supported by FFT diffraction patterns for $CaV_{0.6}Mo_{0.4}O_{3-\delta}$ sample (region 2); Figure S5: Real–resolution HRTEM image supported by FFT diffraction patterns for $CaV_{0.6}Mo_{0.4}O_{3-\delta}$ sample (region 3).

**Author Contributions:** Conceptualization, S.A.B.; methodology, S.A.B. and A.V.K.; validation, S.A.B., E.Y.G. and A.V.K.; formal analysis, S.A.B. and E.Y.G.; investigation, S.A.B. and E.Y.G.; data curation, S.A.B. and A.V.K.; writing—original draft preparation, S.A.B.; writing—review and editing, A.V.K.; visualization, S.A.B.; supervision, S.A.B. All authors have read and agreed to the published version of the manuscript.

**Funding:** This research received no external funding.

**Institutional Review Board Statement:** Not applicable.

**Informed Consent Statement:** Not applicable.

**Data Availability Statement:** Not applicable.

**Acknowledgments:** The work was carried out using the equipment of the Center of Collective Use "Composition of matter" of Institute of High-Temperature Electrochemistry, UB RAS. HRTEM studies were conducted using the equipment of the Center of Collective Use "National Center of Catalyst Research" of the Boreskov Institute of Catalysis, SB RAS. The authors are grateful to Plaksin S.V., Antonov B.D., Khodimchuk A.V. Moskalenko N.I. (all from IHTE UB RAS), and Farlenkov A.S. (Ural Federal University) for their assistance in the experiment.

**Conflicts of Interest:** The authors declare no conflict of interest. The funders had no role in the design of the study; in the collection, analyses, or interpretation of data; in the writing of the manuscript, or in the decision to publish the results.

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
