# Peer review of "The Influence of Oxygen Activity on Phase Composition, Crystal Structure, and Electrical Conductivity of CaV1–xMoxO3±δ"

_crystals, doi:10.3390/cryst12030419_

Round 1
Reviewer 1 Report
This manuscript presents results about the influence of oxygen activity on different properties of perovskite-like CaV1-xMoxO3-δ samples in a range of Mo concentration around x=0.4.
The manuscript is well written, with a linear structure, a clear introduction, detailed methods and a solid discussion of the results.
Only minor correction, in my opinion, are needed to accept the manuscript for publication.
Here a list of my comments/questions for the Authors:
- In the introduction, a detailed background regarding the vanadates is provided. The same attention has not been reserved for the molibdates, with a much shorted and less detailed description.
- Also in the introduction, the importance of a study of the effect of oxygen activity on the properties of these materials is not well discussed.
- At the beginning of the materials characterization the authors mention the presence of impurities and in particular the presence of metallic Mo in the sample. They state that such impurities should not affect the properties of the material but they don't present anything to support this statement. Moreover, later in the text, the presence of metallic Mo and other phases are used to explain the behavior of conductivity and chemical expansion of the material.
- XRD results reported in Figure 1 show the presence of metallic Mo only in samples with x=0.5 and x=0.35. Can the authors comment on this?
- Figure 3 doesn't add great value to the article and I suggest for it to be moved to the supplementary information
- Starting from Figure 4, the discussion is only focused on the comparison between x=0.4 and x=0.5. I strongly suggest to add to the discussion (and especially to the HRTEM part) at least a point below 0.4, since the main statement in the article is that 0.4 differ from the rest of the Mo concentration due to the peculiar nanostructure.
- Regarding the SEM measurements the authors mention that EDX on the samples was not possible/reliable. It is not clear why is it not possible and EDX on a large scale showing an average concentration for the Mo of 0.4 would support the claim that x=0.4 sample is characterized by different phases only at the nanoscale
- Presenting the HRTEM results the authors mention segregation of Ca on the surface of the samples, in the form of Ca oxides. Shouldn't this be evident also from the XRD measurements?
- Comparing the HRTEM measurements on the samples with x=0.4 and 0.5 the authors also report the interplanar spacing. I am puzzled about why the interplanar space for region 2 of the sample with x=0.4 is not the same as the one of the sample with x=0.5, since region 2 has a concentration of Mo of around 0.5
- The discussion about the conductivity is well structured and supported from literature but focus only on the case of x=0.5. The unusual results obtained for x=0.4 are presented without proposing a valid and solid argumentation.
- In figure 8 the authors show and discuss the EDX measurements of samples after oxidation. The same kind of measurements were indicated as not reliable when performed on fresh samples (SEM in Figure 4)
- As several other measurements, also the chemical expansion is reported only for samples with x=0.5 and 0.4 (Figure 9). This is particularly odd, since the chemical expansion is discussed together with the conductivity and figure 6 show the conductivity also for x=0.3.
- In the study of the oxidation process, reported in Figure 10, two different values of the oxygen activity have been used for the two samples. Is there a specific reason for this?
Author Response
We thank the Reviewer for valuable comments.
Please see the attachment.

Reviewer 2 Report
The article is devoted to the study of perovskite-like solid solutions CaV1-xMoxO3-δ with metal-like conductivity near the concentration of Mo equal to x = 0.4. Undoubtedly, the results presented by the authors are of high scientific novelty and practical significance, and are also promising for practical research. In general, the presented results of the study can be accepted for publication after the authors provide answers to all the questions raised by the reviewer during the reading of the article.
1. In the abstract, the authors need to more clearly state the purpose and relevance of this work.
2. The authors should explain and provide more details regarding the synthesis of structures, as well as the choice of sintering conditions and subsequent processing.
3. The authors should present the results of how exactly the phase composition and structural parameters were determined, and why a change in the concentration of the components leads to an increase in the parameters of the crystal lattice.
4. Authors should provide size charts of spherical grains that were used as the objects under study.
5. The results of the study of the chemical expansion of the oxidation process require additional explanation.
Author Response

(The authors gave the same response as above.)

Round 2
Reviewer 1 Report
I thank the authors for their efforts in answering my questions and comments.
While modifications to the manuscript have been minimal, I consider author's answers to all my questions clear and adequate.
Therefore, in my opinion, the manuscript can be accepted for publication.
Reviewer 2 Report
The authors answered all the questions, the article can be accepted for publication.